# Comprehensive Genomic Analysis of Uropathogenic *E. coli*: Virulence Factors, Antimicrobial Resistance, and Mobile Genetic Elements

**DOI:** 10.3390/pathogens13090794

**Published:** 2024-09-13

**Authors:** Kidon Sung, Mohamed Nawaz, Miseon Park, Jungwhan Chon, Saeed A. Khan, Khulud Alotaibi, Ashraf A. Khan

**Affiliations:** 1Division of Microbiology, National Center for Toxicological Research, U.S. Food and Drug Administration, Jefferson, AR 72079, USA; mnawaz023@gmail.com (M.N.); miseon.park@fda.hhs.gov (M.P.); saeed.khan@fda.hhs.gov (S.A.K.); kholoudsafar@gmail.com (K.A.); ashraf.khan@fda.hhs.gov (A.A.K.); 2Department of Companion Animal Health, Inje University, Gimhae 50834, Republic of Korea; alvarmar@naver.com

**Keywords:** whole-genome sequencing, uropathogen, *E. coli*

## Abstract

Our whole-genome sequencing analysis of sixteen uropathogenic *E. coli* isolates revealed a concerning picture of multidrug resistance and potentially virulent bacteria. All isolates belonged to four distinct clonal groups, with the highly prevalent ST131 lineage being associated with extensive antibiotic resistance and virulence factors. Notably, all isolates exhibited multidrug resistance, with some resistant to as many as 12 antibiotics. Fluoroquinolone resistance stemmed primarily from efflux pumps and mutations in gyrase and topoisomerase genes. Additionally, we identified genes encoding resistance to extended-spectrum cephalosporins, trimethoprim/sulfamethoxazole, and various heavy metals. The presence of diverse plasmids and phages suggests the potential for horizontal gene transfer and the dissemination of virulence factors. All isolates harbored genomic islands containing virulence factors associated with adhesion, biofilm formation, and invasion. Genes essential for iron acquisition, flagella biosynthesis, secretion systems, and toxin production were also prevalent. Adding further complexity to understanding the isolates’ genetic makeup, we identified CRISPR-Cas systems. This study underscores the need for continued genomic surveillance in understanding the pathogenic mechanisms and resistance profiles of uropathogenic *E. coli* to aid in developing targeted therapeutic strategies.

## 1. Introduction

Urinary tract infections (UTIs) are the most common outpatient infections in the United States, resulting in an estimated 10.5 million physician visits [1]. *Escherichia coli* stands as the primary microorganism associated with UTIs, accounting for over 70% of cases, followed by *Proteus mirabilis*, *Klebsiella pneumoniae*, *Enterococcus faecalis*, and *Staphylococcus saprophyticus* [2].

UTIs are highly prevalent, affecting at least 12% of men and 50–60% of women during their lifetimes [3]. Even more people experience asymptomatic bacteriuria, in which bacteria are present in the urine without causing symptoms. Commonly used antibiotics for treating UTIs include fluoroquinolones, nitrofurantoin, fosfomycin, trimethoprim-sulfamethoxazole (TMP-SMX), and β-lactams [4]. The emergence of multidrug-resistant (MDR) uropathogenic *E. coli* (UPEC) has become a growing challenge in UTI treatment [5]. Fluoroquinolone resistance rates in Europe have reached 22%, and in the USA, approximately 31% of hospitalized patients carry fluoroquinolone-resistant UPEC [6]. The prevalence of MDR extended-spectrum β-lactamase (ESBL)-producing *E. coli* strains is on the rise, with ESBL-*E. coli* in bacteriuria increasing from 17 to 24% between 2014 and 2020 [7]. Additionally, most ESBL-positive *E. coli* strains often showed combined resistance to fluoroquinolones [7].

UPEC possesses a range of virulence factors, enabling it to colonize the urinary tract and evade host defenses. Adhesins, including type 1 fimbriae (*fim*), P fimbriae (*pap*), and S fimbriae (*sfa*), play a crucial role in establishing infection by recognizing receptors on the uroepithelium [8]. UPEC can burrow into the uroepithelium, releasing toxins such as hemolysin (*hly*), and cytotoxic necrotizing factor 1 (*cnf1*), to damage cells for nutrients and deploy siderophores to steal iron for growth [9]. Evading the immune system, UPEC multiplies, travels to the kidneys, reattaches with adhesins, and unleashes toxins to colonize and damage renal tissue [8].

Whole-genome sequencing (WGS) has revolutionized the study of bacterial pathogens. Unlike traditional tests, WGS provides a detailed blueprint of a pathogen’s genes, revealing its antibiotic resistance mechanisms, mobile genetic elements (MGEs), virulence factors, multilocus sequence typing, and even its serotype [10]. This powerful tool has been used in recent studies to predict antibiotic susceptibility in various pathogens [10,11,12]. In this study, we leveraged WGS to analyze both the antibiotic resistance and virulence determinants of UPEC.

## 2. Materials and Methods

### 2.1. Acquisition of Uropathogenic E. coli

Sixteen *E. coli* strains were obtained from the Department of Veteran’s Affairs, Minneapolis, MN, and stored at −70 °C in a cryogenic preserving medium with ceramic beads using CRYOBANK tubes (Copan, Murrieta, CA, USA). These strains were then revitalized through overnight growth at 37 °C on trypticase soy agar (TSA) plates supplemented with 5% sheep blood (Thermo Fisher Scientific, Waltham, MA, USA).

### 2.2. Determination of Antibiotic Susceptibility

A disk-diffusion assay was used according to established criteria from the Clinical and Laboratory Standards Institute on the following antibiotics: aminoglycosides, β-lactams, (e.g., penicillins, cephalosporins, and carbapenems), chloramphenicol, fluoroquinolones, tetracyclines, a macrolide, fosfomycin, trimethoprim/sulfamethoxazole, and polymyxin B [13].

### 2.3. Effect of Phenylalanine-Arginine β-Naphthylamide (PaβN) on Ciprofloxacin Minimum Inhibitory Concentration (MIC) Using the Broth Microdilution Method

We evaluated the MIC of ciprofloxacin alone and with the efflux inhibitor PaβN (MilliporeSigma, St Louis, MO, USA). Mueller–Hinton broth (Thermo Fisher Scientific) was prepared with ciprofloxacin concentrations ranging from 0.125 to 256 μg/mL in a 96-well plate (Thermo Fisher Scientific). PaβN was added to designated wells to a final concentration of 100 μg/mL. A bacterial suspension of 0.5 McFarland turbidity was added to each well and incubated overnight in a Synergy 2 Multi-Mode Microplate Reader (BIOTEK Instruments, Winooski, VT, USA) at 37 °C with continuous shaking. Bacterial growth was monitored by reading absorbance of 600 nm every 30 min for 24 h. The MIC of ciprofloxacin was determined as the lowest concentration that completely inhibited bacterial growth.

### 2.4. Quantitation of the Efflux Pump and Outer Membrane Gene Expression via Quantitative Reverse Transcription–Polymerase Chain Reaction (qRT-PCR)

The quantification of efflux pumps via qRT-PCR involved the extraction of total RNA from ciprofloxacin-sensitive (CFT008) and -resistant uropathogenic *E. coli* isolates using the RNeasy Protect Bacteria Mini Kit (QIAGEN, Germantown, MD, USA). The reverse transcription of one µg of total RNA was carried out using the SuperScript III First-Strand Synthesis SuperMix (Thermo Fisher Scientific) for qRT-PCR. Quantitative PCR was performed on a CFX 96 Touch Real Time PCR machine (Bio-Rad, Hercules, CA, USA). Each gene (including efflux pumps *acrA*, *acrB*, *ompF*, *norE*, and *mdfA*; with reference to *gapA*, Appendix A) underwent reactions in a total volume of 20 µL of a reaction mix containing 10 µL of Power SYBR GREEN PCR Master Mix (Thermo Fisher Scientific), 50 ng of cDNA, and 1 µL of each primer (final concentration: 300 nM). All reactions were performed in triplicate. The relative amount of the target transcript, known as the relative quantity, was determined through a comparison with the corresponding standard curve, calculated using this equation: relative quantity = 10^([Ct-intercept]/slope)^. For each sample, the relative quantity was then normalized to that of the reference transcript *gapA,* leading to a normalized relative quantity calculated as follows: normalized relative quantity = relative quantity_target transcript_/relative quantity*_gapA_*. The relative expression of each target gene was then calibrated against the corresponding expression using a ciprofloxacin-susceptible *E. coli* strain UTI-2 (MIC 0.016 µg/mL), which served as the control.

### 2.5. Whole-Genome Sequencing

*E. coli* cultures were grown overnight, after which we extracted their genomic DNA using the DNeasy Blood and Tissue Kit (QIAGEN). Next, DNA libraries were prepared for sequencing with the Nextera XT library prep kit (Illumina, San Diego, CA, USA). WGS was performed on an Illumina MiSeq platform, generating paired-end reads of 251 base pairs each. The raw sequence data were processed using CLC Genomics Workbench 21.0.4 (QIAGEN) for trimming and assembly. These sequencing data are publicly available at the National Center for Biotechnology Information (NCBI) BioProject under accession number PRJNA669151 [14].

### 2.6. Identification of Multilocus Sequence Types (MLST), Core Genome MLST (cgMLST), Serotype, FimH and FumC Type, Antibiotic Resistance, Mobile Genetic Elements (MGEs), Phages, CRISPR-Cas Systems, and Virulence

The genetic relatedness and serotypes of the *E. coli* strains were determined using MLST 2.0, cgMLSTFinder 1.2, and SerotypeFinder 2.0, available on the Center for Genomic Epidemiology (CGE) server [15]. Additionally, CHTyper 1.0 was used to identify the types of FimH and FumC proteins [15]. We constructed a phylogenetic tree to visualize the evolutionary relationships among the *E. coli* strains. The Bacterial Genome Tree tool on the Bacterial and Viral Bioinformatics Resource Center (BV-BRC) platform was used for tree generation, and interactive Tree Of Life (iTOL) v5 was used for editing [16,17].

Multiple databases were employed to identify genes and mutations associated with antibiotic resistance in the *E. coli* isolates. These databases included the Comprehensive Antibiotic Resistance Database (CARD), Pathosystems Resource Integration Center (PATRIC), the National Database of Antibiotic Resistant Organisms (NDARO), ResFinderPlus, VRprofile2, and Staramr [18,19,20,21,22]. MGEs, which can facilitate the spread of antimicrobial resistance genes, were identified using Mobile Element Finder v1.0.3 (CGE), VRprofile2, Staramr, and mobileOG-db v1.1.3 of Proksee [21,22,23,24]. Additionally, PHASTER was employed to detect phages [25], while the Clustered Regularly Interspaced Short Palindromic Repeats (CRISPR)-Cas system was characterized using CRISPRCasFinder [26]. VRprofile2 was also used to identify genomic islands (GIs) [22]. The Virulence Factors of Pathogenic Bacteria Database (VFDB), the PATRIC curated virulence database (PATRIC_VF), and the Victors database were used to identify virulence factors in the *E. coli* genomes [27,28,29]. Default settings were used for all software unless otherwise specified.

## 3. Results

### 3.1. Phylogenetic, MLST, cgMLST, Serotype, and CH (fimH/fumC) Type Analysis

Analysis of the phylogenetic tree of the UPEC strains revealed the presence of three major phylogroups (Figure 1). The analysis of serotype, MLST, and CH typing revealed distinct patterns among the UPEC isolates (Table 1). The most prominent finding was the predominance of the ST131 lineage, which included 10 of the 16 isolates. This clonal group was strongly associated with O25 serotype, as all ST131 isolates in this study displayed the O25 serogroup and H4 flagellar type. Within the ST131 isolates, there was consistency in CH typing results. Most isolates exhibited *fumC*40 and *fimH*30 alleles, which are characteristic of the ST131 lineage.

### 3.2. Phenotypic Antimicrobial Susceptibility

We evaluated the susceptibility of our UPEC isolates to a range of antibiotics (Table 2). These antibiotics covered various classes, including aminoglycosides, β-lactams, quinolones, macrolides, tetracyclines, and others. While some strains showed susceptibility to streptomycin, imipenem, chloramphenicol, and fosfomycin, we observed a worryingly high prevalence of resistance to several antibiotics. Most isolates showed resistance to gentamicin, kanamycin, ampicillin, cefepime, nalidixic acid, tetracycline, and polymyxin B. Additionally, the MIC for ciprofloxacin exceeded 32 µg/mL for all UPEC isolates. Significantly, all UPEC strains exhibited MDR, with resistance to at least six different antibiotics. Strains 3385 and 4410, each resistant to 12 antibiotics, had the highest level of resistance.

To investigate the potential involvement of efflux pumps in reducing ciprofloxacin susceptibility among the UPEC strains, we employed the efflux pump inhibitor PAβN to assess efflux activity. Nine UPEC strains exhibited a two- to four-fold decrease in ciprofloxacin MIC when PAβN was present (Table 3). Notably, the ciprofloxacin MIC of three isolates (3887, 4410, 4715) displayed a four-fold reduction in the presence of PAβN, highlighting the role of efflux pumps in diminishing ciprofloxacin susceptibility.

### 3.3. Analysis of Quantification of RNA Expression by qRT-PCR

Table 3 also shows the expression levels of various efflux pump genes in all UPEC isolates. Notably, all strains exhibited the overexpression of at least one efflux pump gene, suggesting a potential role for efflux pumps in antibiotic resistance. Among the individual efflux pumps, *acrA* and *acrB* were the most prevalent, with overexpression detected in 15 isolates. Additionally, the overexpression of *mdfA* and *norE* was seen in 11 and 15 isolates, respectively. Interestingly, 11 isolates showed overexpression of all four efflux pumps tested (*acrA*, *acrB*, *mdfA*, and *norE*). Yet, despite the observed overexpression of efflux pump genes, we found no significant correlation between efflux pump activity and ciprofloxacin MICs. Furthermore, we detected a decreased expression of *ompF* in 15 isolates.

### 3.4. Genotypic Antimicrobial Analysis of UPEC WGS

WGS revealed a concerning prevalence of resistance genes in the UPEC isolates. Notably, a significant portion (11 of 16) harbored genes for resistance against at least one aminoglycoside (Figure 2). This resistance was particularly pronounced in seven isolates, which carried combinations more than three aminoglycoside resistance genes. Isolate UPEC 3385 stood out with the highest number of these resistance genes. However, some isolates defied this trend. Five (2250, 2624, 3636, 4169, and 4331) lacked detectable genes for any of the tested aminoglycosides.

All UPEC isolates, except 3604, carried at least one β-lactam resistance gene. The most concerning examples were 3385 and 3575, which displayed a combination of three different β-lactam resistance genes: *bla*_CTX-M-14_, *bla*_CTX-M-15_, and *bla*_OXA-1_, alongside *bla*_TEM-1B_. Additionally, two chloramphenicol-resistant genes, *catB3* and *clmA1*, were identified in two UPEC isolates.

The *mph(A)* gene, conferring low-level macrolide resistance, was found in eight isolates, while six isolates carried the quaternary ammonium compound-resistant gene, *qacEΔ1*. The hydrogen peroxide-resistant gene, *sitABCD*, was nearly ubiquitous among all isolates. Isolate 3385 harbored both copper-resistance genes, *pcoA-D* and *pcoR-S*, as well as silver-resistance genes, *silA-C*, *silE*, *silP*, and *silR-S*. Mercury-resistance genes, including *merA*, *merC-E*, *merP*, *merR*, and *merT*, were found in three isolates.

Many efflux pump genes were present in all UPEC isolates. Isolate 3385 was found to contain the *qnrB1* gene, a plasmid-mediated gene associated with low-level resistance to quinolones. Additionally, tetracycline resistance genes *tet(A)* and *tet(B)* were identified in four and three isolates, respectively. Among the UPEC isolates, seven carried dihydrofolate reductase genes *dfrA*, including variants *dfrA14* and *dfrA17*, encoding enzymes conferring trimethoprim resistance. Furthermore, seven isolates harbored *sul1* genes, while four contained *sul2* genes, both of which encode sulfonamide resistance.

Thirteen isolates had amino acid substitution in PmrB: 123 (Glu → Asp) associated with chromosome-mediated colistin resistance. Point mutations related to fosfomycin resistance were detected in CyaA: 352 (Ser → Thr), GlpT: 448 (Glu → Lys), PtsI: 25 (Val → Ile), and UhpT: 350 (Glu → Gln), with varying frequencies among the isolates. Moreover, amino acid substitutions associated with quinolones were observed in GyrA, ParC, ParE, and MarR among all isolates, with specific mutations such as Ser83 → Leu and Asp87 → Asn in GyrA and Ser80 → Ile in ParC universally present.

### 3.5. Analysis of MGEs, Insertion Elements (ISs), Phages, and CRISPR-Cas Systems

WGS unveiled a variety of plasmid replicon types among the UPEC isolates (Table 4). Col(BS512) and IncFIA replicon types were the most frequently identified, being present in all isolates. Each UPEC isolate harbored at least four different replicon types, indicating the prevalence of multiple plasmids within these strains. Additionally, Col156 and IncFIB replicon types were found in 15 isolates, highlighting their common occurrence across the UPEC strains. Conversely, replicon types IncP1, IncX1, IncI1-I(α), and IncQ1 were each identified in only one isolate.

An analysis of transposons revealed several types circulating within the UPEC isolates, with Tn*3* and Tn*5403* being the most abundant. The presence of additional transposons, such as Tn*20* and Tn*3000*, emphasized the dynamic nature of the UPEC genome and the potential for the mobilization of genetic material. All UPEC isolates harbored MITEEc1, a miniature inverted transposable element belonging to the IS630 family, and an enterobacterial repetitive intergenic consensus (ERIC) sequence, potentially associated with mRNA stability [30].

MGEs in the UPEC isolates carried a formidable arsenal of antibiotic resistance genes. This included genes for resistance against aminoglycosides, (*aac(3)-IIa*, *aadA1*, *aadA5*, *ant(2″)-Ia*, *ant(3″)-Ia*, *aph(3″)-Ib*, and *aph(6)-Id*), β-lactams, (*bla_CTX-M-14_*, *bla_CTX-M-15_*, and *bla_TEM-1B_*), chloramphenicol (*cmlA1*), macrolides (*mph(A)*), quaternary ammonium compounds (*qacEΔ1*), quinolones (*qnrB1*), tetracyclines, (*tet(A)* and *tet(B)*), and trimethoprim and sulfonamides (*dfrA14*, *dfrA17*, *sul1*, and *sul2*).

Whole-genome sequencing revealed a surprising abundance and diversity of IS elements within the UPEC isolates (Figure 3). Sixty-five ISs were identified in this study. All isolates harbored a remarkable number, exceeding 18 distinct IS elements each, suggesting a dynamic interplay of MGEs within these strains. Three isolates, 2284, 3887, and 4410, stood out with the most complex profiles, containing a staggering 29 ISs apiece. IS629 and ISSfl10 were remarkably consistent, having been identified in all UPEC isolates. Additionally, IS26 and ISEc1 were found in 15 isolates, followed closely by IS2 and ISEc27, present in 14 isolates.

Phages may bolster antibiotic resistance, environmental resilience, and host-cell adhesion, potentially enhancing pathogenicity. Upon genome analysis for phages, we observed that each UPEC isolate contained a minimum of five phages, with isolate 3575 exhibiting the highest count of 16 phages (Appendix A). Shiga toxin 1-converting bacteriophage BP-4795 (PHAGE_Entero_BP_4795_NC_004813) emerged as the most prevalent, identified in 14 isolates. Following BP-4795 in prevalence was *Escherichia* phage P88 (PHAGE_Entero_P88_NC_026014), found in 13 isolates. *Escherichia* phage DE3 (PHAGE_Entero_DE3_NC_042057) was detected in 12 isolates, while *Pectobacterium* phage ZF40 (PHAGE_Pectob_ZF40_NC_019522) was present in 10 isolates. Several phages capable of carrying Shiga toxins (stxA1 and stxB1), including Shiga toxin 1-converting bacteriophage BP-4795 (PHAGE_Entero_BP_4795_NC_004813), *Escherichia* phage SH2026Stx1 (PHAGE_Escher_SH2026Stx1_NC_049919), Stx2-converting phage Stx2a_F451 (PHAGE_Stx2_c_Stx2a_F451_NC_049924), *Enterobacteria* phage VT2phi_272 (PHAGE_Entero_VT2phi_272_NC_028656), *Enterobacteria* phage YYZ-2008 (PHAGE_Entero_YYZ_2008_NC_011356), and *Enterobacteria* phage phiP27 (PHAGE_Entero_phiP27_NC_003356) were detected.

In addition to *Escherichia* phages, two *Shigella* phages capable of carrying Shiga toxins, *Shigella* phage 75/02 Stx (PHAGE_Shigel_Stx_NC_029120) and *Shigella* phage POCJ13 (PHAGE_Shigel_POCJ13_NC_025434), were identified. Furthermore, a phage capable of carrying cytolethal distending toxins, *Enterobacteria* phage fiAA91-ss (PHAGE_Entero_fiAA91_ss_NC_022750), was detected. Phages from *Salmonella*, *Bacillus*, *Burkholderia*, and *Pseudomonas* were also identified.

Bacteria possess a remarkable immune system, known as “clustered regularly interspaced short palindromic repeats-CRISPR associated” (CRISPR-Cas). This system acts like a shield, protecting bacteria from foreign genetic elements like viruses and plasmids. CRISPR loci were examined in 16 UPEC isolates, revealing a total of 57 confirmed loci (Table 5). Notably, isolate 4410 showed a robust defense mechanism, harboring seven CRISPR loci, while others had at least two. The spacer sequences within each CRISPR locus ranged from 1 to 11, with repeat lengths spanning 23 to 54 bp. The average size of spacers varied between 32 and 60.5 bp, while the overall CRISPR length ranged from 82 to 699 bp. I-E was the most prevalent CRISPR-Cas system subtype observed in the UPEC strains. 

Across the strains, 21 distinct consensus-repeat sequences were identified. Notably, the most common sequences we encountered, CGACCCCCACCATGTCAAGGTGGTGCTCTAACCAACTGAGCTA and AGTTCACTGCCGTACAGGCAGCT, were present in nine and seven isolates, respectively. In the CRISPR-Cas system, the *cas1* and *cas2* genes are crucial components located proximal to the CRISPR locus, essential for its functionality. Hence, their presence was investigated across all CRISPR loci. The analysis revealed that only three UPEC strains—2250, 3887, and 4410—contained both core *cas* genes, while the remaining 13 strains lacked these genes. Subsequently, the *cas* genes present in these three strains were further characterized (Figure 4). Additionally, CRISPR-related proteins near CRISPR included endonuclease Cas1, integrase Cas2, helicase Cas3, and proteins Cse1, Cse2, Cas7, Cas5, and Cas6.

All UPEC isolates harbored at least two GIs (Appendix A). Notably, isolates 4410 and 4715 exhibited the highest count, each harboring 13 genomic islands. Interestingly, none of the isolates carried genes for antibiotic resistance. However, except for isolates 2842, 3079, and 3887, all isolates possessed virulence genes. These genes were associated with various virulence factors, including type I fimbriae (*fimA-C*, *fimE*, and *fimI*), *E. coli* common pilus (ECP) operon (*ecpA-E* and *ecpR*), curli biogenesis (*csgA-G*), capsule biogenesis (*kpsC-F*, *kpsS-T*, and *kpsU*), type III secretion system (*espL1* and *spaP-Q*), and hemolysin toxins (*hlyA-B* and *hylC-D*).

The UPEC isolates displayed a wide array of virulence genes implicated in adherence, invasion, iron uptake, secretion systems, and toxin production (Figure 5). Notably, isolate 2250 exhibited the highest number of virulence genes, followed by 4410 and 2284. Adherence-associated genes were ubiquitous among all UPEC isolates, including those encoding type 1 fimbriae (*fimA-C*, *fimE*, and *fimI*), the *E. coli* common pilus (ECP) operon (*ecpA-E* and *ecpR*), and genes implicated in biofilm formation. However, we observed some variation in the presence of genes for curli biogenesis (*csgA-G*) and type P fimbriae (*papA-K* and *papX*). An analysis of UPEC isolates revealed the presence of several virulence factors associated with invasion. All isolates harbored genes for *aslA*, *ompA*, and *kpsD*. Capsular antigens belonging to group 2 (K1, K5, and K52) were also identified. Interestingly, isolate 4410 stood out in possessing the gene *invH*, which encodes an invasion protein that plays a critical role in breaching host-cell defenses.

We found a remarkably high prevalence of 36 genes associated with iron acquisition. In fact, all isolates possessed every one of these genes except for three: *chuT*, *shuT*, and *ybtE*. Also, we identified seven genes (*flgC*, *flgG-H*, *fliA*, *fliG*, *fliM*, and *fliP*) essential for flagella production, with all genes present in every UPEC isolate except for *fliA*. An analysis of secretion system genes, involved in transporting various molecules in and out of bacteria, revealed contrasting distribution patterns. Genes associated with the Type II secretion system (*gspC-J*, *gspK-M*) were found in nearly all isolates. Conversely, genes linked to the Type III secretion system specifically linked to the locus of enterocyte effacement (LEE) (*espL4*, *espR1*, *espR4*, *espX1*, *espX4-5*, *espY1*, and *espY2-4*) were identified exclusively in three isolates (2250, 3887, and 4410). While a secreted autotransporter toxin (*sat*) was present in all UPEC isolates, genes encoding other toxins, including vacuolating autotransporter toxin (*vat*), enterotoxin (*senB*), cytotoxic necrotizing factor 1 (*cnf1*), and hemolysins (*hlyA-D*, *hlyE*), exhibited a more sporadic distribution pattern.

## 4. Discussion

This comprehensive analysis of UPEC isolates using WGS paints a concerning picture of MDR and potentially virulent bacteria. MLST, serotype, and CH typing analyses revealed four distinct groups within the UPEC isolates studied. Notably, the most prevalent types belonged to ST131, O25:H4, and fumC40/fimH30. The predominance of ST131 among the UPEC isolates aligns with previous reports, as it is recognized as the most prevalent *E. coli* lineage associated with extraintestinal pathogenic *E. coli* (ExPEC) infections worldwide [31]. This clonal complex is particularly problematic due to its association with both virulence and antimicrobial resistance. ST131 is an MDR clone commonly associated with resistance to fluoroquinolones, extended-spectrum β-lactams, and carbapenems [32]. Beyond its resistance profile, ST131 is considered a particularly virulent pathogen due to its extensive arsenal of virulence-associated genes, such as *papG* (P fimbrial adhesins), *fimH* (type 1 fimbriae), *kpsM II* (group 2 capsule synthesis), *iucD* (aerobactin), *iutA* (aerobactin receptor), *fyuA* (siderophore yersiniabactin receptor), *sat* (secreted autotransporter toxin), *hlyA* (α-hemolysin), and *cnf1* (cytotoxic necrotizing factor 1) [33]. These genes encode factors that enhance bacteria’s ability to adhere to host cells, evade the immune system, and cause tissue damage.

The CH typing analysis of our UPEC isolates revealed variability in both *fumC* and *fimh* alleles, highlighting genetic diversity within the strains. The most common *fumC* allele was *fumC*40, found in the majority of isolates (10 out of 16). For *fimh*, *fimh*30 was the most prevalent allele, identified in 10 isolates, while *fimh*64 and *fimh*29 were present in fewer isolates. The variability in *fimh* alleles, alongside *fumC* typing, underscores the importance of these genes in UPEC-strain differentiation and their role in pathogenicity. The identification of these specific alleles correlates with known UPEC clonal groups, such as ST131, which often harbor *fimh*30 [34]. The diversity in *fimh* and *fumC* typing among our isolates provides insights into their potential virulence.

This study also delved into the mechanisms underpinning the reduced susceptibility of UPEC isolates to fluoroquinolones. By employing PAβN, we uncovered the involvement of efflux pumps in diminishing ciprofloxacin susceptibility in a subset of UPEC strains. Nine isolates exhibited a two-to-four-fold decrease in ciprofloxacin MIC when co-treated with PAβN. Furthermore, our analysis identified multiple-point mutations in *gyrA*, *parC*, *parE*, and *marR* genes, which likely serve as the primary mechanism driving ciprofloxacin resistance in the UPEC isolates. Intriguingly, we also noted the overexpression of efflux pump genes *acrA-B*, *mdfA*, *norE*, and *ompF* across the isolates. This observation suggests that efflux pump overexpression acts synergistically with mutations in gyrase and topoisomerase genes, further contributing to the reduced susceptibility of UPEC strains to ciprofloxacin.

The WGS of UPEC isolates revealed a concerning prevalence of key antibiotic resistance genes, particularly *bla*_CTX-M-14_, *bla*_CTX-M-15_, *bla*_OXA-1_, and *bla*_TEM-1B_. These genes encode resistance to extended-spectrum cephalosporins, traditionally used to treat UTIs. This finding aligns with observations by Ruppe et al., who reported similar β-lactamase genes, (*bla*_CTX-M-15_, *bla*_OXA-1_, and *bla*_TEM-1B_), in *E. coli* isolated from community-acquired UTIs in Cambodia [35]. This extensive β-lactam resistance significantly limits the treatment options available for UTIs caused by these specific strains.

This study also identified resistance to trimethoprim-sulfamethoxazole (TMP-SMX) among the analyzed UPEC isolates. SMX is a commonly used antibiotic for treating UTIs, and this finding highlights a significant threat to this critical therapy’s effectiveness. WGS revealed two key resistance mechanisms targeting SMX. First, seven isolates harbored dihydrofolate reductase genes (*dfrA*), specifically variants *dfrA14* and *dfrA17*. Second, seven isolates contained *sul1* genes, while four contained *sul2* genes. Additionally, recent investigations have confirmed the presence of *dfrA1* and *sul1* in UPEC isolates [36,37], providing further evidence of the widespread dissemination of these resistance genes and the resulting threat to the efficacy of SMX in UTI treatment.

The widespread presence of efflux pump genes in all UPEC isolates underscores their significance in mediating multidrug resistance. The diverse repertoire of these genes identified in our UPEC isolates suggests a multifaceted mechanism through which these bacteria evade the effects of antibiotics. Among the efflux pump genes identified, *acrAB-tolC*, *acrAD-tolC*, and *acrEF-tolC* are members of the resistance–nodulation–cell division family of efflux pumps, known for their broad substrate specificity and ability to extrude a wide range of antibiotics, including β-lactams, fluoroquinolones, and tetracyclines [38]. Additionally, *mdtABC-tolC*, *mdtEF-tolC*, and *mdtM* belong to the multidrug and toxic-compound extrusion family of efflux pumps, involved in the extrusion of various antimicrobial compounds [39]. The presence of such efflux pump genes as *mexAB-oprM* is particularly noteworthy due to their role in conferring resistance to multiple classes of antibiotics, including β-lactams, fluoroquinolones, and macrolides [40]. These efflux pumps contribute to UPEC’s adaptive response to antibiotic exposure and can confer resistance to a broad array of antibiotics.

Some isolates displayed resistance mechanisms against several heavy metals. Notably, isolate 3385 emerged as a standout case, harboring genes for resistance to both copper (*pcoA-D* and *pcoR-S*) and silver (*silA-C*, *silE*, *silP*, and *silR-S*). Additionally, three isolates harbored genes for mercury resistance (*merA*, *merC-E*, *merP*, *merR*, and *merT*). The identification of resistance genes targeting heavy metals raises concerns about the potential for UPEC strains to persist in environments with these agents. Copper and silver have antimicrobial properties and are sometimes incorporated into medical devices or wound dressings [41]. The presence of resistance mechanisms against these metals could potentially limit the effectiveness of these strategies for preventing or controlling UPEC infections. Moreover, the co-occurrence of metal resistance genes with antibiotic resistance genes highlights the potential for co-selection, wherein exposure to one type of stressor may inadvertently promote the emergence or dissemination of antibiotic resistance.

Our exploration of plasmid replicon types among the UPEC isolates unveiled a diverse range of replicons, indicating the likely presence of multiple plasmids per isolate. The most prevalent replicon types, Col(BS512) and IncFIA, were consistently detected across all isolates. IncFIA is particularly significant, given its association with conjugative plasmids capable of transferring antibiotic resistance genes among bacteria [42]. Notably, IncF plasmids identified in UPEC ST405 have been implicated in disseminating various antibiotic resistance determinants, including ESBLs and quinolone resistance genes, thereby contributing to the global spread of antibiotic resistance [43].

The presence of Col(BS512) is noteworthy. Known as the *Shigella boydii* plasmid pBS512 with replicon type FIIA, it is categorized as an invasive plasmid with relation to the type III secretion system [44]. Additionally, the identification of replicons such as IncP1, IncX1, IncI1-I(α), and IncQ1, each found in only one isolate, presents an intriguing aspect. These replicons are less commonly encountered in UPEC, implying potential acquisition events from diverse sources or the presence of unique genetic elements not typically associated with UPEC plasmids.

To our knowledge, this study represents the first comprehensive exploration of phage diversity in UPEC. Our analysis uncovered a wide repertoire of bacteriophages among UPEC isolates, suggesting the potential for phage-mediated genetic exchange and dissemination of virulence factors within these strains. Notably, Shiga toxin 1-converting bacteriophage BP-4795 emerged as the most prevalent, detected in 14 isolates. This phage is known to encode a type III effector that can be translocated into human cells through the locus of an enterocyte effacement-encoded type III secretion system [45]. We identified several phages capable of carrying Shiga toxins, including Shiga toxin 1-converting bacteriophage BP-4795, *Escherichia* phage SH2026Stx1, Stx2-converting phage Stx2a_F451, *Enterobacteria* phage VT2phi_272, *Enterobacteria* phage YYZ-2008, and *Enterobacteria* phage phiP27. The presence of these phages in UPEC isolates suggests a potential risk factor for the emergence of highly virulent strains capable of causing severe complications. Additionally, the detection of *Enterobacteria* phage fiAA91-ss, a carrier of cytolethal distending toxin genes, further underscores the complexity of virulence potential encoded via these phages.

The identification of CRISPR-Cas systems in our UPEC isolates sheds light on the potential importance of this adaptive immune system in protecting against foreign genetic elements. We observed variability in spacer-sequence length and composition within CRISPR loci, highlighting diversity among UPEC strains. Our findings on CRISPR-Cas subtype prevalence aligned with previous reports, with the type I-E system being the most dominant [46]. Typically, strains with active CRISPR-Cas systems would be expected to have fewer virulence genes [47]. Lindenstrauss et al. found an inverse correlation between the presence of CRISPR-Cas systems and acquired virulence determinants in *Enterococcus faecalis* [48]. However, our study presents a contrasting scenario. We identified three UPEC strains (2250, 3887, and 4410) harboring functional type I-E CRISPR-Cas systems, yet these strains also carried a higher number of virulence genes compared to other isolates. This intriguing finding suggests that, in these specific UPEC strains, the presence of type I-E CRISPR-Cas systems might not be directly linked to the acquisition of virulence genes. Adding another layer of complexity, Dang et al. [49] observed that UPEC was less likely than non-pathogenic strains to possess CRISPR loci. This suggests a potential role for CRISPR in acquiring MGEs of *E. coli*, which may not always translate to virulence genes. These contrasting observations highlight the intricate relationship between CRISPR-Cas systems and virulence factors in UPEC.

The detection of GIs in our UPEC isolates provides significant insights into the genomic architecture and potential virulence of these strains. Our study revealed that all UPEC isolates harbored at least two GIs, with isolates 4410 and 4715 each containing the highest number, 13 GIs. Interestingly, despite the extensive presence of GIs, none of the isolates carried genes for antibiotic resistance within these regions. This finding contrasts with some reports in which GIs have been shown to frequently harbor antibiotic resistance genes, contributing to the MDR phenotype in pathogenic *E. coli* [50]. The absence of such genes in our isolates’ GIs might indicate a different evolutionary path or selective pressure acting on these UPEC strains, focusing more on virulence than on antibiotic resistance. Except for isolates 2842, 3079, and 3887, all isolates possessed virulence genes within their GIs. These genes were associated with various virulence factors, such as adhesion, biofilm formation, type III secretion system, and hemolysin production; such findings are consistent with GIs’ known role in harboring virulence determinants that enhance *E. coli*’s pathogenicity [51].

Our study underscores the critical role of adherence-associated virulence genes in UPEC pathogenesis. These genes orchestrate the initial attachment of bacteria to host tissues and subsequent biofilm formation, establishing and perpetuating UTIs [52]. Among the identified genes, those encoding type 1 fimbriae (*fimA-C*, *fimE*, and *fimI*) and the *E. coli* common pilus operon (*ecpA-E* and *ecpR*) were universally present in all UPEC isolates. Type 1 fimbriae are well-established facilitators of UPEC adherence to bladder epithelial cells, promoting colonization and infection [52]. Similarly, the *E. coli* common pilus operon enhances UPEC’s adhesive and invasive capabilities during interaction with the host bladder [53]. This ubiquity aligns with other reports, including those by Mirzahosseini et al., who identified *fimA*, *fimH*, and *ecpA* as the most common virulence genes in UPEC isolates worldwide [54]. Interestingly, some variability was observed in the presence of genes associated with curli biogenesis (*csgA-G*) and type P fimbriae (*papA-K* and *papX*). Curli are functional amyloid fibers that contribute to biofilm formation and host colonization, while type P fimbriae are involved in adhesion to specific receptors on host cells [52]. This variability in adherence-associated genes highlights the potential for diverse virulence strategies among UPEC isolates.

Beyond the initial attachment to host cells, some UPEC strains possess additional virulence factors that enable them to invade deeper tissues, further complicating UTIs. All isolates were found to harbor the genes *ompA* and *aslA*, crucial for bacterial invasion and survival within host tissues. Furthermore, all isolates harbored capsular antigens belonging to group 2, specifically K1, K5, and K52. These capsular antigens provide resistance to phagocytosis via immune cells and serum bactericidal activity, facilitating the establishment and persistence of UTIs [55]. Of particular interest is isolate 4410, which notably possessed the gene *invH*. This gene is associated with the ability of bacteria to invade epithelial cells, an essential step in the pathogenesis of invasive infections [56]. The presence of *invH* in this isolate suggests that isolate 4410 may have enhanced invasive capabilities compared to other isolates, potentially influencing its pathogenicity and clinical outcomes.

Iron is an essential nutrient for bacterial growth and virulence, particularly in the iron-limited environment of the urinary tract. A remarkable finding was the high prevalence of genes associated with iron acquisition—36—with all isolates possessing all but three (*chuT*, *shuT*, and *ybtE*). This finding is consistent with previous reports that highlight the importance of iron-acquisition systems in UPEC pathogenicity. Iron-acquisition systems, such as siderophores and heme-uptake mechanisms, allow UPEC to thrive in the host by scavenging iron from host proteins [57].

Flagella are essential for motility and colonization, contributing to UPEC’s ability to ascend the urinary tract and establish infection. Our study identified seven genes essential for flagella production (*flgC*, *flgG-H*, *fliA*, *fliG*, *fliM*, and *fliP*), with all genes being present in every UPEC isolate except for *fliA*. This gene profile resembles that in the work of Lane et al. [58], who reported that flagellar motility plays a significant role in UPEC pathogenesis. The presence of these genes in nearly all isolates suggests that motility is a key factor in the persistence and virulence of UPEC.

An analysis of secretion system genes revealed contrasting distribution patterns. Genes associated with the Type II secretion system (*gspC-J* and *gspK-M*) were found in nearly all isolates. The Type II secretion system is known to be involved in the secretion of a variety of enzymes and toxins that contribute to bacterial virulence [59]. In contrast, genes linked to the Type III secretion system, specifically those associated with the locus of enterocyte effacement (LEE) (*espL4*, *espR1*, *espR4*, *espX1*, *espX4-5*, *espY1*, and *espY2-4*), were identified exclusively in three isolates (2250, 3887, and 4410). The LEE-encoded Type III secretion system is typically associated with enteropathogenic and enterohemorrhagic *E. coli* strains, in which it plays a crucial role in delivering effector proteins into host cells, subverting host-cell signaling and immune responses [60].

The toxin genes among UPEC isolates vary significantly in their potential to cause host-tissue damage. Notably, the *sat* gene was present in all UPEC isolates. Encoding a serine protease autotransporter, it is one of the most frequent virulence genes found in UPEC [61]. In contrast, genes encoding other toxins, such as *vat*, *senB*, *cnf1*, *hlyA-D*, and *hlyE*, exhibited more sporadic distribution patterns. This finding aligns with observations by Kim et al., who identified similar toxins in *E. coli* isolates from hospitalized UTI patients [62]. Interestingly, we found *sat* to be universally present; Kim et al. reported the *hlyE* gene in all isolates. This variation in toxin gene profiles suggests that, while all UPEC isolates possess a baseline level of virulence potential due to the ubiquitous presence of *sat*, the additional virulence factors they harbor may be considerably diverse.

Earlier research has highlighted significant variability in the presence of virulence factors across UPEC isolates, with specific genes like *eae*, *pap*, and *afa* showing a lower prevalence in certain populations. For example, the *eae* gene was present in only one of 225 UPEC strains, suggesting that not all virulence factors common in other pathogenic *E. coli* strains are equally prevalent in UPEC [63]. Another study reported that none of their UPEC isolates contained the *pap* GI gene [64]. Blanco et al. further demonstrated the low prevalence of the *afa* gene among UPEC isolates. Their study detected *afa* in only a small subset of isolates from patients with cystitis (4.3%) and asymptomatic bacteriuria (2.4%), highlighting the variability in the presence of this adhesin gene among UPEC strains [65]. These findings underscore that the presence of virulence factors is not uniform across all UPEC isolates, and their distribution can vary significantly based on the genetic background and environmental pressures influencing each strain. This evidence supports the notion that, while certain virulence genes may be prevalent, others can show much lower frequencies, contributing to the observed diversity in virulence potential among UPEC isolates.

## 5. Conclusions

Our comprehensive analysis of UPEC isolates through WGS paints a concerning picture of MDR and potentially virulent UPEC strains. Widespread resistance to various antibiotics, including fluoroquinolones, extended-spectrum cephalosporins, and TMP-SMX, poses a significant challenge in treating UTIs. The diverse mechanisms of resistance, including efflux pumps, mutations, and plasmid-mediated gene transfer, highlight the adaptability of these bacteria. Furthermore, the presence of virulence factors associated with adhesion, invasion, iron acquisition, and toxin production underscores the potential of these UPEC strains to cause severe UTIs. This study emphasizes the necessity of continued genomic surveillance in understanding UPEC’s pathogenic mechanisms and resistance profiles, thus aiding in the development of targeted therapeutic strategies.

## Figures and Tables

**Figure 1 pathogens-13-00794-f001:**
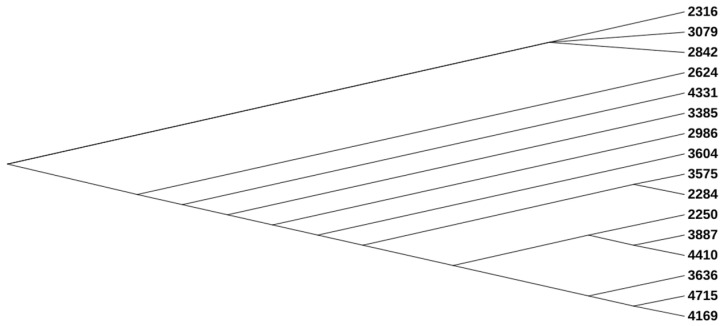
Phylogenetic tree of the UPEC isolates.

**Figure 2 pathogens-13-00794-f002:**
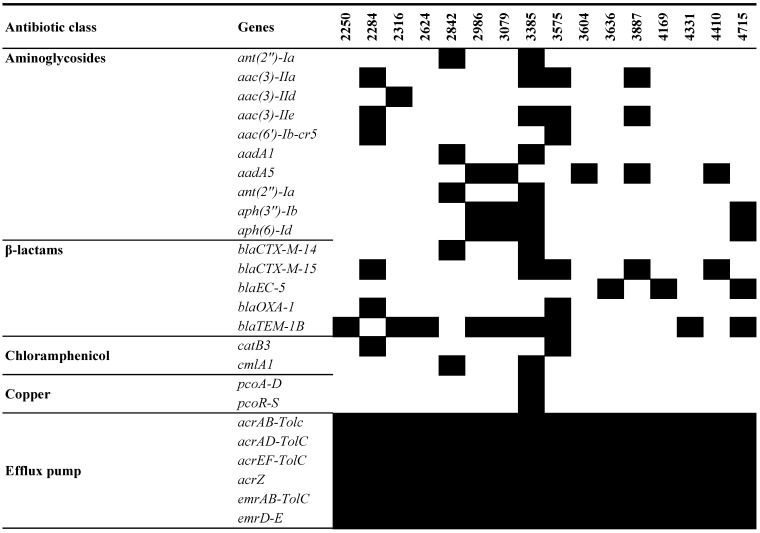
Antimicrobial resistance determinants in the UPEC isolates. The figure shows the presence (black for antimicrobial resistance gene) or absence (blank) of antimicrobial resistance genes.

**Figure 3 pathogens-13-00794-f003:**
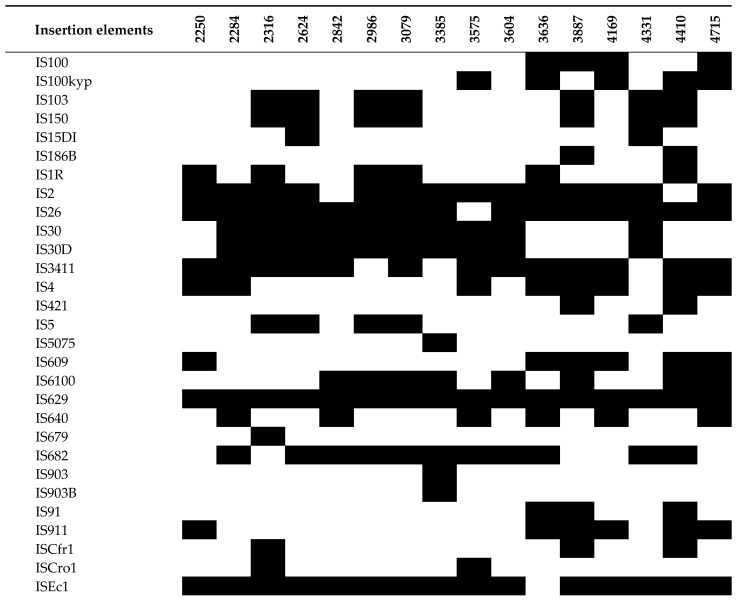
Insertion elements (ISs) in the UPEC isolates. The figure shows the presence (black for antimicrobial resistance gene) or absence (blank) of antimicrobial resistance genes.

**Figure 4 pathogens-13-00794-f004:**
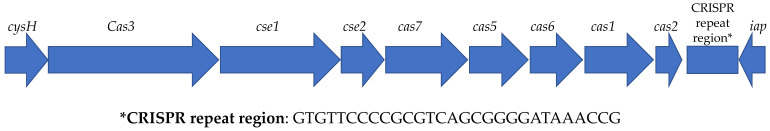
Schematic view of CRISPR-Cas in UPEC isolate 2250.

**Figure 5 pathogens-13-00794-f005:**
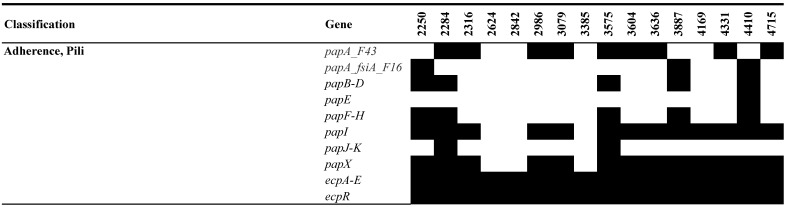
Virulence determinants in the UPEC isolates. The figure shows the presence (black) or absence (blank) of virulence genes.

**Table 1 pathogens-13-00794-t001:** MLST, cgMLST, serotype, CH type analysis of the UPEC isolates.

		2250	2284	2316	2624	2842	2986	3079	3385	3575	3604	3636	3887	4169	4331	4410	4715
**Multilocus sequence typing**	** *adk* **	18	53	53	53	53	53	53	53	53	53	14	35	14	53	35	14
	** *fumC* **	106	40	40	40	40	40	40	40	40	40	14	37	14	40	37	14
	** *gyrB* **	17	47	47	47	47	47	47	47	47	47	10	29	10	47	29	10
	** *icd* **	6	13	13	13	13	13	13	13	13	13	200	25	200	13	25	200
	** *mdh* **	5	36	36	36	36	36	36	36	36	36	17	4	17	36	4	17
	** *purA* **	5	28	28	28	28	28	28	28	28	28	7	5	7	28	5	7
	** *recA* **	4	29	29	29	29	29	29	29	29	29	10	73	10	29	73	10
	**ST**	**393**	**131**	**131**	**131**	**131**	**131**	**131**	**131**	**131**	**131**	**1193**	**405**	**1193**	**131**	**405**	**1193**
**cgMLST**	**cgST**	152,922	152,915	152,921	152,912	152,913	152,924	152,918	152,917	152,920	152,911	72,154	152,916	152,923	152,914	158,995	152,919
**Sero typing**	**O type**	O25	O25	O25	O25	O25	O25	O25	O25	O25	O25	O75	O102	O75	O25	O102	O75
	**H type**	H1	H4	H4	H4	H4	H4	H4	H4	H4	H4	H5	H6	H5	H4	H6	H5
**CH typing**	**FumC**	106	40	40	40	40	40	40	40	40	40	14	37	14	40	37	14
	**FimH**	54	30	30	30	30	30	30	30	30	30	64	29	64	30	29	64

**Table 2 pathogens-13-00794-t002:** Frequency of phenotypic antibiotic resistance among sixteen UPEC isolates.

	Resistant	Intermediate	Susceptible
**AZR**	7	2	7
**GEN**	15	1	0
**KAN**	10	5	1
**STR**	4	0	12
**AMP**	16	0	0
**CEF**	16	0	0
**CFP**	6	4	6
**IMI**	5	4	7
**CHL**	0	0	16
**NAL**	16	0	0
**TET**	13	3	1
**DOX**	6	4	6
**FOS**	0	0	16
**TMP-SMX**	7	1	8
**POL**	15	0	1

AZR: azithromycin, GEN: gentamicin, KAN: kanamycin, STR: streptomycin, AMP: ampicillin, CEF: cefepime, CFP: cefoperazone, IMI: imipenem, CHL: chloramphenicol, NAL: nalidixic acid, TET: tetracycline, DOX: doxycycline, FOS: fosfomycin, TMP-SMX: trimethoprim/sulfamethoxazole, POL: polymyxin B.

**Table 3 pathogens-13-00794-t003:** Ciprofloxacin MIC with and without the inhibitor PaβN and gene expression of efflux pump genes and outer membrane proteins in the UPEC isolates.

			Overexpression of Efflux Pump Genes				Alterations of Outer Membrane Protein
Strains	Ciprofloxacin	Ciprofloxacin + PAβN (100 ug/mL)	*acrA*	*acrB*	*mdfA*	*norE*	*ompF*
**2250**	32	32	3.18	4.97	14.61	4.59	2.24
**2284**	256	128	1.90	3.20	2.91	3.09	−1.43
**2316**	256	256	2.93	2.85	3.28	3.77	−1.29
**2624**	64	64	1.62	1.34	−1.65	1.76	−2.75
**2842**	64	64	1.50	1.94	3.42	3.12	−1.95
**2986**	256	128	1.71	1.89	−1.35	2.12	−2.61
**3079**	128	64	1.35	3.01	1.56	1.98	−2.17
**3385**	256	128	1.70	1.43	−1.67	1.58	−2.77
**3575**	256	128	−1.16	−1.17	−1.48	1.53	−2.58
**3604**	128	128	2.28	2.69	1.86	3.43	−2.73
**3636**	256	128	2.83	4.29	4.33	2.43	−1.07
**3887**	256	64	3.54	4.92	4.85	4.02	−14.61
**4169**	256	256	3.09	4.18	2.88	3.03	−1.40
**4331**	256	256	2.12	1.76	1.74	2.57	−2.09
**4410**	128	32	3.91	4.06	1.18	2.08	−3.64
**4715**	256	64	1.18	1.32	−2.45	−1.57	−3.18

**Table 4 pathogens-13-00794-t004:** Mobile genetic elements carrying antibiotic resistance genes in the UPEC isolates. **Bold**: common.

Strains	Plasmid	Composite Transposons	Transposons	Miniature Inverted Repeats	Antibiotic Resistance Genes
**2250**	**Col(BS512)**, Col156, **IncFIA**, IncFIB, IncP1	cn_5813_IS911	TnpA_Tn3, TnpR_Tn3	**MITEEc1**	* blaTEM-1B *
**2284**	**Col(BS512)**, Col156, **IncFIA**, IncFIB, IncFII	cn_4876_IS629, cn_15494_IS629	Tn5403, TnpA_Tn5403, TnpA_TnAs1, TnpR_Tn5403	**MITEEc1**	*aac(3)-Ila*, *tet(A)*
**2316**	**Col(BS512)**, Col(MG828), Col156, **IncFIA**, IncFIB, IncFII			**MITEEc1**	
**2624**	**Col(BS512)**, Col(MG828), Col156, **IncFIA**, IncFII		TnpA_Tn3	**MITEEc1**	
**2842**	**Col(BS512)**, Col156, **IncFIA**, IncFIB, IncFII			**MITEEc1**	*ant(2″)-Ia*, *ant(3″)-Ia*, *blaCTX-M-14*, *cmlA1*, *mph(A)*, *qacE*, *sul1*
**2986**	**Col(BS512)**, Col156, **IncFIA**, IncFIB, IncFII			**MITEEc1**	*aadA5*, *dfrA17*, *mph(A)*, *qacE*, *sul1*
**3079**	**Col(BS512)**, Col(MG828), Col156, **IncFIA**, IncFIB, IncFII			**MITEEc1**	*aadA5*, *aph(3″)-Ib*, *aph(6)-Id*, *dfrA17*, *mph(A)*, *qacE*, *sul2*, *tet(A)*
**3385**	**Col(BS512)**, Col156, **IncFIA**, IncFIB(K), IncFII		Tn5403, TnpA_ISPa38, TnpA_Tn3, TnpA_Tn3000, TnpA_Tn5403, TnpR_Tn3, TnpR_Tn5403	**MITEEc1**	*aadA1*, *ant(2″)-Ia*, *aph(3″)-Ib*, *aph(6)-Id*, *blaCTX-M-15*, *blaTEM-1B*, *cmlA1*, *qacE*, *qnrB1*, *sul2*
**3575**	**Col(BS512)**, Col(MG828), Col156, **IncFIA**, IncFIB, IncFII, IncX1		Tn5403, TnpA_Tn5403, TnpR_Tn5403	**MITEEc1**	
**3604**	**Col(BS512)**, **IncFIA**, IncFIB, IncFII		Tn5403, TnpA_Tn5403, TnpR_Tn20, TnpR_Tn5403	**MITEEc1**	*aadA5*, *dfrA17*, *mph(A)*, *qacE*, *sul1*
**3636**	**Col(BS512)**, Col156, **IncFIA**, IncFIB, IncI1-I(α)			**MITEEc1**	
**3887**	**Col(BS512)**, Col(MG828), Col156, IncFIA, IncFIB, IncFII	cn_3324_ISEc1	TnpA_Tn3	**MITEEc1**	*aac(3)-Ila*, *aadA5*, *blaCTX-M-15*, *dfrA17*, *mph(A)*, *qacE*, *sul1*
**4169**	**Col(BS512)**, Col156, **IncFIA**, IncFIB	cn_1487_IS629, cn_1794_IS26, cn_8908_IS911, cn_8908_ISEc31		**MITEEc1**	
**4331**	**Col(BS512)**, Col156, **IncFIA**, IncFIB, IncFII			**MITEEc1**	* blaTEM-1B *
**4410**	**Col(BS512)**, Col156, **IncFIA**, IncFIB	cn_3324_ISEc1, cn_5813_IS911	TnpA_Tn3, TnpR_Tn3	**MITEEc1**	*aadA5*, *blaCTX-M-15*, *dfrA17*, *mph(A)*, *qacE*, *sul1*
**4715**	**Col(BS512)**, Col(MG828), Col156, **IncFIA**, IncFIB, IncQ1			**MITEEc1**	*dfrA14*, *tet(B)*

**Table 5 pathogens-13-00794-t005:** CRISPR-Cas system in the UPEC isolates.

Strains	Type	Cas Genes	CRISPR Consensus Repeat	Numbers of Spacer	Repeat Length	Mean Size of Spacers	CRISPR Length
**2250**	I, I-E	*cas3*, *cse1*, *cse2*, *cas7*, *cas5*, *cas6*, *cas1*, *cas2*	CGCGTCTTATCAGGCCTACGAGTTCGGTGCTGTGTAGGTCGGATAAGGCGTTCA	1	54	42	149
			GAGTTCCCCGCGCCAGCGGGGATAAACCG	3	29	32	211
			GTGTTCCCCGCGTCAGCGGGGATAAACCG	11	29	32	699
			CGACCCCCACCATGTCAAGGTGGTGCTCTAACCAACTGAGCTA	1	43	38	123
			GTTCACTGCCGTACAGGCAGCTTAGAAA	3	28	32	207
**2284**	ND	*ND*	GCCGGATGCGGCGTGAACGCCTTATCCGGCCTACAAAAGAAATGCAG	1	47	48	141
			AGTTCACTGCCGTACAGGCAGCT	1	23	37	82
			CGACCCCCACCATGTCAAGGTGGTGCTCTAACCAACTGAGCTA	1	43	38	123
**2316**	I	*cas3*	AGTTCACTGCCGTACAGGCAGCT	1	23	37	82
			AATGCCTGATGCGACGCTTGTCGCGTCTTATCATGCCTACAAGT	1	44	57	144
**2624**	I	*cas3*	GCCGGATGCGGCGTGAACGCCTTATCCGGCCTACAAAAGAAATGCAG	1	47	48	141
			CCACCTTTTTTACCTGCTTCAGATGC	1	26	40	91
			AGTTCACTGCCGTACAGGCAGCT	1	23	37	82
**2842**	I	*cas3*	CGACCCCCACCATGTCAAGGTGGTGCTCTAACCAACTGAGCTA	1	43	38	123
			ATCTGCCTGTACGGCAGTGAACT	1	23	37	82
**2986**	ND	*ND*	CCACCTTTTTTACCTGCTTCAGATGC	1	26	40	91
			ATCTGCCTGTACGGCAGTGAACT	1	23	37	82
**3079**	I	*cas3*	CGACCCCCACCATGTCAAGGTGGTGCTCTAACCAACTGAGCTA	1	43	38	123
			AGTTCACTGCCGTACAGGCAGCT	1	23	37	82
			GCCGGATGCGGCGTGAACGCCTTATCCGGCCTACAAAAGAAATGCAG	1	47	48	141
**3385**	ND	*ND*	GCCGGATGCGGCGTGAACGCCTTATCCGGCCTACAAAAGAAATGCAG	1	47	48	141
			AGTTCACTGCCGTACAGGCAGCT	1	23	37	82
			CGACCCCCACCATGTCAAGGTGGTGCTCTAACCAACTGAGCTA	1	43	38	123
**3575**	I	*cas3*	GCCGGATGCGGCGTGAACGCCTTATCCGGCCTACAAAAGAAATGCAG	1	47	48	141
			AGTTCACTGCCGTACAGGCAGCT	1	23	37	82
			CCACCTTTTTTACCTGCTTCAGATGC	1	26	40	91
**3604**	ND	*ND*	CGACCCCCACCATGTCAAGGTGGTGCTCTAACCAACTGAGCTA	1	43	38	123
			GCCGGATGCGGCGTGAACGCCTTATCCGGCCTACAAAAGAAATGCAG	1	47	48	141
			AGTTCACTGCCGTACAGGCAGCT	1	23	37	82
**3636**	I	*cas3*	GTTCACTGCCGTACAGGCAGCTTAGAAA	2	28	32	147
			CCGAGCCGTAGGCCGGATAAGGCGTTCACGC	1	31	56	117
			CGACCCCCACCATGTCAAGGTGGTGCTCTAACCAACTGAGCTA	1	43	38	123
			TGAACGCCTTATCCGACCTACACAGCACTGAACTCGTAGGCCTGATAAGACGCG	1	54	42	149
**3887**	I, I-E	*cas3*, *cse1*, *cse2*, *cas7*, *cas5*, *cas6*, *cas1*, *cas2*	TGAACGCCTTATCCGACCTACACAGCACTGAACTCGTAGGCCTGATAAGACGCG	1	54	42	149
			GTGTTCCCCGCGCCAGCGGGGATAAA	9	26	35	574
			CCACCTTTTTTACCTGCTTCAGATGC	1	26	40	91
			CAGCGTCGCATCAGGCATTGTGCACGATTGCCGGATGCGGCGTGAACGCCTT	1	52	47	150
**4169**	I	*cas3*	TTTCTAAGCTGCCTGTACGGCAGTGAAC	2	28	32	147
			ACGCTGCCGCGTCTTATCGGGCCTACAAAAGTTCTGAACCGT	1	42	49	132
			TTGATTGCCGGATGCGGCACGAGTGCCTTATCCGGCCTAC	2	40	60.5	240
			CGCGTCTTATCAGGCCTACGAGTTCGGTGCTGTGTAGGTCGGATAAGGCGTTCA	1	54	42	149
			CCGAGCCGTAGGCCGGATAAGGCGTTCACGC	1	31	56	117
			CGACCCCCACCATGTCAAGGTGGTGCTCTAACCAACTGAGCTA	1	43	38	123
**4331**	ND	*ND*	CCACCTTTTTTACCTGCTTCAGATGC	1	26	40	91
			ATCTGCCTGTACGGCAGTGAACT	1	23	37	82
**4410**	I, I-E	*cas3*, *cse1*, *cse2*, *cas7*, *cas5*, *cas6*, *cas1*, *cas2*	AAGGCGTTCACGCCGCATCCGGCAATCGTGCATAATGCCTGATGCGACGCTG	1	52	47	150
			CCACCTTTTTTACCTGCTTCAGATGC	1	26	40	91
			GTGTTCCCCGCGCCAGCGGGGATAAA	9	26	35	574
			TGAACGCCTTATCCGACCTACACAGCACTGAACTCGTAGGCCTGATAAGACGCG	1	54	42	149
			CGACCCCCACCATGTCAAGGTGGTGCTCTAACCAACTGAGCTA	1	43	38	123
			CCCAATTAAGTGAAGAGCAGGTGACGAAGTTACTGCATCGCAAA	1	44	58	145
			GAGTTCCCCGCGCCAGCGGGGATAAACCG	10	29	32	638
**4715**	ND	*ND*	TTTGTAGGCCTGATAAGACGCGCCAGCGTCGCATCAGGC	1	39	49	126
			GTTCACTGCCGTACAGGCAGCTTAGAAA	2	28	32	147
			GTAGGCCGGATAAGGCACTTGTGCCGCATCCGGCA	1	35	57	126
			TGAACGCCTTATCCGACCTACACAGCACTGAACTCGTAGGCCTGATAAGACGCG	1	54	42	149
			ACGCTGCCGCGTCTTATCGGGCCTACAAAAGTTCTGAACCGT	1	42	49	132

## Data Availability

The original contributions presented in the study are included in the article/Appendix A, further inquiries can be directed to the corresponding author.

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
