# Peer review of "Comprehensive Genomic Analysis of Uropathogenic E. coli: Virulence Factors, Antimicrobial Resistance, and Mobile Genetic Elements"

_pathogens, 2024, doi:10.3390/pathogens13090794_

Round 1
Reviewer 1 Report
Comments and Suggestions for Authors
This paper describes the whole genome sequence analysis of uropathogenic E. coli strains, compares the occurrence of anti-microbial resistance and virulence and correlates these to genome components. It gives a comprehensive description of the variety of resistances found in the different UPEC isolates, the numbers and types of genomic elements, e.g., insertion elements, bacteriophages, pathogenicity islands, etc., and the virulence associated genes present in the various isolates. In the discussion, the authors discuss the significance of the isolates possessing combinations of all/ some of these genomic elements and genes in terms of evolution and the selection pressure that can act on them.
The study essentially illustrates the fact that pathogenic isolates’ arsenals are usually subsets/ combinations, no isolate possesses all the identified virulence factors or drug resistance mechanisms. This is not a novel concept. All this information is relevant and important in combatting urinary tract infections by E. coli. How is this information being used currently? Are any of the genes discussed potential drug targets? How can this information be potentially used? What is the source of these pathogenic isolates? Does this correlate to the genomic elements they possess? If the authors could shed some light on these issues, the paper would be of wider interest.
Reviewer 2 Report
Comments and Suggestions for Authors
The authors present the complete genome analysis of 16 strains of uropathogenic E. coli (UPEC) to understand the virulence and resistance of UPEC. The analysis of the genome and the mobile genetic elements allowed a broad understanding of the genetic load associated with virulence and antibiotic resistance, its frequency in UPEC strains, its location within the genome and/or mobile genetic elements.
The writing and analysis of the sequences was adequate; however, it could be improved by clarifying some points.
Abstract
It is important to point out the number of UPEC strains analyzed and what is the conclusion of the work.
Materials and methods
In the section on the acquisition of uropathogenic E. coli, it is important to show which inclusion criteria were used to select the UPEC strains, patient data such as sex, age, and whether they were isolated from recurrent UTI. What are the keys to each strain.
Why a more exhaustive analysis of ciprofloxacin resistance?
Results
A better analysis of the results is required. In the description of the results shown in Table 1, it is also observed that there is a relationship between the serotype, ST, CH typing
I consider that Table 2 could show only the frequency of resistance to each antibiotic evaluated, as described in the text.
Lines 183 and 184 are the same text.
Line 193 should be removed from the text
Improve the description of Table 4. How many plasmids and how many transposons were identified.
Fig. 3, Fig 5, Fig 7 If the genetic map is not described, I do not think it is relevant to show it in the article.
Tables 5 and 7 are very large, it could be passed as a supplementary table. In addition, the phages corresponding to each strain should be indicated with a dividing line.
Reviewer 3 Report
Comments and Suggestions for Authors
The paper by Sung et al. describes the whole genome sequencing (WGS) analysis of a number of clinical isolates from the Department of Veteran's Affairs in Minneapolis, MN. By using WGS, a direct side by side comparison of the isolates was made. Some things to do to improve the paper are noted below.
1. State up front how many isolates were examined. Are these recent isolates or have they been in the freezer a whole? The limited number isolates makes it hard to note trends.
2. You should simplify the determination of antibiotic susceptibility. Just state a disk diffusion assay was used according to established criteria from the Clinical Laboratory Standards Institute on the following antibiotics: x,y,z (13). Get rid of reference 13 and make 14 reference 13. What control strain did you use?
3. Your qRT-PCR analysis is missing some essential information. What other controls did you use? How many replicates were done?
4. Line 117 FumC is not an adhesin protein.
5. Present Table 1 as landscape and not portrait so everything fits in one line.
6. Table 2 is incomplete. In the table legend, state how many UPEC isolates were examined. All of the abbreviations for the drugs needs to positioned as a footnote at the bottom. Add another footnote that tells the reader was R, I and S mean.
7. Table 3 What is considered overexpression? Tell the reader the fold differences in transcription rather than the vague + and -.
8. Prokaryotic genes have the first three letters lowercase, so it should be oprM, opmH, cmr, and tolC.
9. One isolate carried the invA gene, which is interesting. Does this gene have homology with the invA gene of Salmonella or Yersinia? If it is Salmonella, where are the other inv genes that are part the type III secretion system?
10. The discussion has a section with a different font size.
11. Some emphasis on the the variability of the fimH gene in each isolate should be noted since one typing scheme uses fimH and fumC.
12. Your discussion at times is vague and does not usually make not of some older literature. It is not surprising that the flagellar genes are present since several papers have noted that flagella are an important virulence factor of UPEC. Older literature has noted the lower frequencies of specific virulence factor genes.
Round 2
Reviewer 3 Report
Comments and Suggestions for Authors
The authors have addressed my concerns.